# Kidney Transplant Outcomes after Prolonged Delayed Graft Function

**DOI:** 10.3390/jcm11061535

**Published:** 2022-03-11

**Authors:** Cullan V. Donnelly, Maria Keller, Liise Kayler

**Affiliations:** 1Jacobs School of Medicine and Biomedical Sciences, SUNY-University at Buffalo, 955 Main Street, Buffalo, NY 14203, USA; cullando@buffalo.edu; 2Department of Surgery, SUNY-University at Buffalo, 100 High Street, Buffalo, NY 14203, USA; mkeller1@ecmc.edu; 3Transplant and Kidney Care Regional Center of Excellence, Erie County Medical Center, 462 Grider Street, Buffalo, NY 14215, USA

**Keywords:** kidney transplant, allograft function, delayed graft function

## Abstract

Background: The protracted recovery of renal function may be an actionable marker of post-transplant adverse events, but a paucity of data are available to determine if the duration of graft recovery serves to stratify risk. Materials and Methods: Single-center data of adult-isolated deceased-donor kidney transplant (KTX) recipients between 1 July 2015 and 31 December 2018 were stratified by delayed graft function (DGF) duration, defined as time to serum creatinine < 3.0 mg/dL. Results: Of 355 kidney transplants, the time to creatinine < 3.0 mg/dL was 0–3 days among 96 cases (DGF ≤ 3), 4–10 days among 85 cases (DGF4-10), 11–20 days among 93 cases (DGF11-20), and ≥21 days for 81 cases (DGF ≥ 21). DGF ≥ 21 recipients were significantly more likely to be male, non-sensitized, and receive kidneys from donors that were older, with donation after circulatory death, non-mandatory share, hypertensive, higher KDPI, higher terminal creatinine, and longer cold and warm ischemia time. On multivariate analysis, DGF ≥ 21 was associated with a 5.73-fold increased odds of 12-month eGFR < 40 mL/min compared to DGF ≤ 3. Lesser degrees of DGF had similar outcomes. Conclusions: Prolonged DGF lasting over 20 days signifies a substantially higher risk for reduced eGFR at 1 year compared to lesser degrees of DGF, thus serving as a threshold indicator of increased risk.

## 1. Introduction

Kidney transplantation increases the quality and length of life for most individuals with end-stage renal disease [1]. However, due to low kidney availability, not everyone on the list receives a transplant, and nearly half die or become too sick while waiting [2,3,4,5]. Transplant programs can maximize kidney availability by accepting a wider range of deceased-donor kidneys, yet nearly 20% of kidneys that are recovered for transplantation are discarded every year [6].

A common reason for kidney discard is ischemia-reperfusion injury caused by events such as cold ischemia time, acute kidney injury, and donation after circulatory death (DCD) [7,8]. Ischemia-reperfusion injury manifests as delayed graft function (DGF) after the transplant. DGF is common, occurring in 15% to 50% of primary deceased-donor kidney transplants [8,9,10,11,12,13], and can reach as high as 93% with DCD kidneys [14]. If DGF recovery is prompt, graft outcomes are generally similar to kidneys that functioned immediately [10]. However, long durations of DGF are associated with poor graft outcomes in most studies [10,11,12,15]. Additionally, DGF is challenging to manage [16,17], and there are no approved medications to prevent or shorten it. Therefore, transplant centers may have concerns using kidneys that are likely to develop protracted DGF. The broadening distribution of deceased-donor kidneys and longer travel distances may raise the frequency and severity of DGF and lead to even more organ discard. 

To optimize kidney acceptance by transplant programs, there is a need to better understand the impact of prolonged DGF on post-transplant outcome. Significantly worse graft survival has been seen with DGF durations exceeding 6 days [11], 8 days [15], 13 days [10], 14 days [18], 15 days [12], 21 days [19], and 29 days [14] compared to shorter durations of DGF [10,11,15] or no DGF [11,12,14,15,18,19,20]. However, limitations of prior studies render it difficult to discern the contribution of ischemia-reperfusion injury to the poorer graft outcomes. Some analyses lacked adjustment for potentially confounding donor quality factors (e.g., age, hypertension) [14,15,18,19,20]. Others included recipients with primary non-function [12,15,18], thereby inflating the poor results of longer DGF groups. The high acute rejection seen in most studies may be due to non-contemporaneous immunosuppression strategies. Only one study was based in the United States (US) [20], limiting generalizability. Lastly, graft function has been infrequently examined as an early outcome.

A current understanding of DGF in the US is needed to improve kidney use and patient outcomes. A clearer understanding may also assist in the development of management strategies and therapeutics to prevent or mitigate DGF. Our center has a high incidence of DGF from importing deceased-donor kidneys. We analyzed our center’s data of adult kidney transplant recipients to examine the association between DGF duration and 1-year estimated glomerular filtration rate. We also examined graft survival and assessed other confounders of graft loss, including acute rejection and BK and cytomegalovirus viremia. 

## 2. Materials and Methods

Single-center data of adult (aged ≥ 18 years) kidney-only transplant recipients of deceased-donor kidneys were obtained between 1 July 2015 and 31 December 2018, totaling 409 cases. Two cases were excluded due to a loss of follow-up. Recipients with grafts that never functioned were also excluded: 6 recipients that died before the determination of graft function and 17 recipients with primary non-function. DGF duration, for the purposes of this study, was defined as the time required for the allograft to reach a serum creatinine level < 3.0 mg/dL and having had a permanent discontinuation of hemodialysis. The use of creatinine as an endpoint enables the inclusion of all patients, including those with functional DGF. Therefore, we excluded 2 recipients who never achieved a serum creatinine below 3.0 mg/dL and 3 recipients with an unknown creatinine trajectory due to server downtime during a cyberattack on the hospitals’ electronic medical records. Lastly, we excluded 29 recipients with a CPRA ≥ 99%, since DGF in highly sensitized recipients may more likely be due to immunologic injury. After exclusions, the final analytic size was 355 subjects. DGF duration was classified into 4 groups stratified by the 25th, 50th, and 75th percentiles: 0–3 days (DGF ≤ 3), 4–10 days (DGF4-10), 11–20 days (DGF11-20), and ≥21 days (DGF ≥ 21). Recipient, donor, and transplant covariates assessed are specified in Table 1 and Table 2.

### 2.1. Endpoints

The primary endpoint was the estimated glomerular filtration rate (eGFR) at 3, 6, and 12 months. eGFR was calculated using the abbreviated Modification of Diet in Renal Disease formula [21]. The creatinine level used in the eGFR equation was the mean of all serum creatinine values between 76 and 104 days, 163 and 197 days, and 330 and 365 days post-KTX, for eGFRs at 3, 6, and 12 months, respectively. 

The secondary endpoints were assessed at 12 months, including: (1) graft survival (defined as a return to dialysis, re-transplantation, allograft nephrectomy, or patient death), (2) patient survival, (3) acute rejection (biopsy-proven and excluding cases with ‘borderline changes’ as defined by Heilman et. al) [22], (4) cytomegalovirus viremia (CMV), and (5) BK polyomavirus viremia (BKV). CMV and BKV were defined as two consecutive serum polymerase chain reaction viral loads greater than 500 international units/mL.

### 2.2. Study Environment

All transplants were ABO-compatible with negative B- and T-cell flow crossmatch or negative virtual crossmatch. During the study period, kidney transplant immunosuppression consisted of anti-thymocyte globulin (*n* = 351) or Basiliximab (Interleukin-2 monoclonal antibody; *n* = 4) induction and oral tacrolimus, mycophenolate mofetil, and corticosteroid therapy. Introduction of the calcineurin inhibitor was at the first day post-transplant. Kidney transplant biopsies during periods of DGF were obtained every 10 to 14 days per protocol. During the study period, cold ischemia time was prolonged primarily due to organ allocation and transportation.

### 2.3. Statistical Analyses

The appropriate functional forms of covariates were determined by exploratory data analysis in unadjusted models and perceived impact on clinical meaningfulness. Univariate associations between exposure groups were examined using the Chi-square test or Fisher’s exact test for categorical variables (presented as number and proportion) and ANOVA for continuous variables whose distributions approximated normality (summarized as mean and standard deviation). Multivariate analyses of binary endpoints were performed using logistic regression to estimate odds ratios (aOR) and 95% confidence intervals (95% CI) for exposure groups after accounting for potential confounders with an alpha of <0.05 associated with the outcome required for entry into the model. All statistical analyses were conducted using the IBM SPSS Statistics 25. All *p*-values were 2-sided, and <0.05 was considered statistically significant. The study was approved by the University at Buffalo Institutional Review Board. 

## 3. Results

Of 355 kidney transplants, the time to creatinine < 3.0 mg/dL was 0–3 days among 96 cases (DGF ≤ 3), 4–10 days among 85 cases (DGF4-10), 11–20 days among 93 cases (DGF11-20), and ≥21 days for 81 cases (DGF ≥ 21). DGF ≥ 21 recipients were significantly more likely to be male, non-sensitized (Table 1), and receive kidneys from donors that were older, with donation after circulatory death, non-mandatory share, hypertensive, and have higher KDPI, higher terminal creatinine, and longer cold and warm ischemia time (Table 2). Within each DGF group, the mean eGFR progressively increased over the 12-month post-transplant period (Figure 1). The largest increase was in the DGF ≥ 21 group, wherein the mean eGFR increased from 39 mL/min/1.73 m^2^ at 3 months to 46 mL/min/1.73 m^2^ at 12 months, a seven-point difference (Figure 1).

Between the DGF groups, the eGFR rate was significantly lower in the DGF ≥ 21 group compared to the other DGF groups throughout the 12-month follow-up period. For example, the eGFR within the DGF ≥ 21 group was 39 to 46 mL/min/1.73 m^2^ throughout the first post-transplant year. In comparison, the eGFR within the DGF0-3 group during the same period was 60 to 65 mL/min/1.73 m^2^ (Figure 1).

On multivariate analysis, DGF ≥ 21 was associated with a 5.73-fold increased odds of 12-month eGFR < 40 mL/min compared to DGF ≤ 3. However, lesser DGF durations of 4–10 days and 11–20 days were not significantly different relative to DGF ≤ 3 in terms of 12-month eGFR < 40 mL/min (Table 3).

### Secondary Outcomes

One-year graft failure of the DGF 0–3, 4–10, 11–20, and ≥21 recipients was 6.3%, 8.2%, 6.5%, and 11.1%, respectively (*p* = 0.617). Although the between-group differences did not reach significance, the 1-year graft failure within the DGF ≥ 21 group was nearly twice that of the DGF0-3 group, and the lack of significance may be due to the low power of the analysis to detect low event rate differences. Rates of 1-year acute rejection ranged between 8.9% and 17.1% and were not significant between groups (*p* = 0.830). CMV (*p* = 0.420) and BK (*p* = 0.839) frequencies were not found to be statistically significant across the DGF groups. There were no significant differences in 12-month patient mortality (*p* = 0.849) (Table 4).

## 4. Discussion

In this single-center retrospective study, including a population with a high incidence of DGF, we found that a prolonged DGF of more than 20 days to reach a serum creatinine <3 mg/dL was independently associated with reduced 1-year eGFR compared to a DGF duration of 0–3 days. The prolonged DGF group also had the lowest 1-year graft survival; however, the difference was not significant, likely due to the small sample size. Importantly, middle-range durations of DGF of 4–10 days and 11–20 days did not have a differential effect on eGFR or graft survival.

Our finding of an association of prolonged DGF and lower 1-year graft function occurred in the setting of low frequencies of other causes of graft injury—acute rejection, CMV, and BK. Therefore, the poorer graft function in the longest DGF group may reflect the severity of ischemia-reperfusion injury. It has been hypothesized that the maladaptive repair of parenchymal and tubular cells affected by acute kidney injury contributes to fibrosis and loss of functioning renal mass [23]. Additionally, the poorer graft function may be due to lower intrinsic kidney quality. Poor donor quality characteristics were more highly concentrated in the prolonged DGF group, and our model may have been incompletely adjusted. In the context of a severe organ shortage, the utilization of kidneys at risk of DGF is an important option to provide the life-extending opportunity that transplant offers [7]. However, given the detrimental effect of prolonged DGF on the clinical course and outcomes, there is a need for interventions to mitigate the effects of DGF. Many treatments have been tested in clinical trials but have shown minimal or no difference in DGF rates with intervention, or no eventual effect on allograft function or survival [10]. We show that patients with prolonged DGF over 3 weeks represent a subset of individuals with a high risk for poor graft function. These patients could be flagged for management strategies (such as facilitating timely re-transplantation) or future therapeutic interventions to reduce or prevent graft functional decline. 

We also found that short durations of DGF yield acceptable early outcomes. The potential incremental effect of DGF duration is incompletely studied because few prior studies examined three or more DGF duration groups [10,12,14,19], and one report is not interpretable due to the inclusion of primary non-function in the longest DGF group [12]. Two studies found similar graft survival between all DGF duration groups with cut-offs up to >21 days [19] and >28 days [14]. In contrast, an Australian registry study found similar death-censored graft loss only with lower DGF durations of 1–4 days and 5–7 days, whereas longer DGF durations up to 35 days were independently associated with low graft survival. Interestingly, the authors also found that only 8.4% of the effect between DGF duration and graft loss was explained by acute rejection, suggesting that the effects of DGF on graft loss were not totally mediated by acute rejection [10]. The graft loss may have resulted from other unfavorable effects of ischemic-reperfusion injury or graft quality, since the only donor characteristics included in the model were age, DCD, and cold ischemia time. Ischemic injury has been thought to heighten acute rejection [8]. Significantly larger acute rejection rates have been noted in prolonged DGF groups [10,11,12,18,19,20] compared to shorter DGF durations; however, these studies were conducted in cohorts nearly 1 to 2 decades ago and had higher frequencies of acute rejection that are not seen with current immunosuppression strategies. Our contemporary findings of the low acute rejection rates and acceptable eGFR amongst recipients experiencing middle-length durations of DGF align with studies showing a lack of differential outcomes between recipients with and without DGF if acute rejection is avoided [8,9,13].

### Study Limitations

There are a number of limitations in this study. Most transplant recipients in the study were Caucasian and Black, limiting the generalizability of our findings. As with other retrospective studies, our analysis may be confounded by variables that were not included in our models, such as blood loss, kidney allograft size, perioperative hypotension [24], proteinuria or nephrotic syndrome, and immunosuppressant choice and level. Our follow-up was only one year. A longer-term follow-up is needed to assess the important endpoint of graft survival. Our definition of DGF duration is not standard; therefore, comparison across studies is difficult. However, there are numerous definitions of DGF and DGF duration, and a time-based analysis of serum creatinine change has been suggested to perform best in being able to predict reduced graft function at one year [25]. We agree with the suggestion that “we are in need of a unifying definition of DGF that likely involves serum and urine biomarkers in order to improve the field of transplant nephrology” [26].

## 5. Conclusions

This study shows a significantly lower estimated glomerular filtration rate (eGFR) 1 year after transplantation in patients with prolonged DGF compared with short or no duration of DGF. We hypothesize that this is caused by low intrinsic donor quality and ischemia-reperfusion injury since the incidence of acute rejection was low overall. The lower graft function was not associated with significantly decreased graft survival, likely due the short follow-up period.

## Figures and Tables

**Figure 1 jcm-11-01535-f001:**
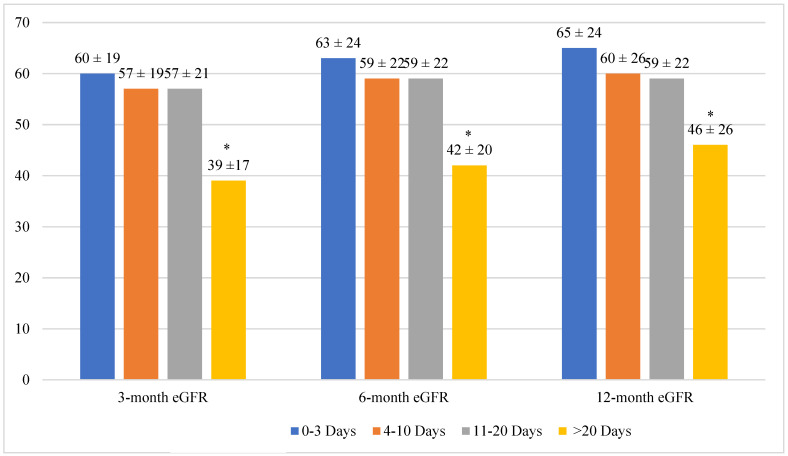
Mean eGFR (mL/min/1.73 m^2^) ± SD at 3, 6, and 12 months Across DGF Duration Groups (* indicates statistical significance, *p* < 0.001).

**Table 1 jcm-11-01535-t001:** Donor characteristics across DGF duration groups (*n* = 355).

Characteristic *n* (%) or Mean ± SD	DGF 0–3 *n* = 96	DGF 4–10 *n* = 85	DGF 11–20 *n* = 93	DGF ≥ 21 *n* = 81	*p* Value
Age, years	36.5 ± 6.4	39.0 ± 15.6	41.0 ± 15.1	46.5 ± 12.9	<0.001
Sex, male	57 (59.4)	58/85 (68.2)	62 (66.7)	50 (61.7)	0.568
Race, black	13 (13.5)	15/85 (17.6)	6 (6.5)	13 (16.0)	0.125
Body Mass Index (kg/m^2^)	27.6 ± 6.4	29.6 ± 9.7	31.1 ± 13.1	30.9 ± 8.8	0.052
Diabetes mellitus	5 (5.2)	7 (8.2)	9 (9.7)	8/81 (9.9)	0.633
Hypertension	18 (18.8)	27 (31.8)	34 (36.6)	34 (42.0)	0.006
Kidney Donor Profile Index	44.0 ± 25.3	54.0 ± 23.3	53.9 ± 22.0	63.7 ± 19.4	<0.001
Terminal Serum Creatinine (mL/min/1.73 m^2^)	1.0 ± 0.5	1.4 ± 1.0	1.8 ± 1.7	1.6 ± 1.4	<0.001
Donor Required Dialysis	1 (1.0)	5 (5.9)	4 (4.3)	1 (1.2)	0.177
Donation after circulatory death Warm ischemia time > 45 min	23 (24.0) 4 (4.2)	24 (28.2) 4 (4.7)	46 (49.5) 5 (5.4)	49 (60.5) 10 (12.3)	<0.001 <0.001
Non-Mandatory Share	26 (27.1)	39 (45.9)	49 (52.7)	58 (71.6)	<0.001
Artery Luminal Narrowing > 50% * Interstitial Fibrosis > 50% *	11 (11.5) 5 (5.2)	11 (12.9) 8 (9.4)	15 (16.1) 9 (9.7)	17 (21.0) 2 (2.5)	0.315 0.177
Pump Terminal resistive index ^1^	0.2 ± 1.3	0.2 ± 0.1	0.2 ± 0.1	0.3 ± 0.1	0.492
Cold Ischemia Time ≥ 30 h	16 (16.7)	32 (37.6)	27 (29.0)	36 (44.4)	<0.001

Non-Mandatory Share, defined as non-mandatory share offer at sequence number > 50; ^1^ Hypothermic machine perfusion. * Confirmed via biopsy.

**Table 2 jcm-11-01535-t002:** Recipient characteristics across DGF duration groups.

Characteristic *n* (%) or Mean *±* SD	DGF 0–3 *n* = 96	DGF 4–10 *n* = 85	DGF 11–20 *n* = 93	DGF ≥ 21 *n* = 81	*p* Value
Age, years	53.6 ± 14.6	55.3 ± 12.3	54.2 ± 12.0	57.4 ± 12.4	0.245
Sex, male	44 (45.8)	46 (54.1)	68 (73.1)	57 (70.4)	<0.001
Race, black	26 (27.1)	27 (31.8)	32 (34.4)	30 (37.0)	0.528
Body Mass Index (kg/m^2^)	28.8 ± 6.1	30.9 ± 6.1	31.5 ± 6.0	30.9 ± 5.7	0.013
Diabetes mellitus	36 (37.5)	35 (41.2)	47 (50.5)	35 (43.2)	0.327
Human Leukocyte Antigen ABDr Mismatch	4.6 ± 1.2	4.3 ± 1.5	4.5 ± 1.3	4.5 ± 1.0	0.501
Dialysis Vintage > 2 years	34 (35.4)	36 (42.4)	42 (45.2)	32 (39.5)	0.568
Preemptive transplant	30 (31.3)	9 (10.6)	14 (15.1)	17 (21.0)	0.003
Calculated panel reactive antibody (cPRA) > 0% ^1^	40 (41.7)	30 (35.3)	23 (24.7)	18 (22.2)	0.015
Estimated post-transplant survival (%)	43.5 ± 28.4	51.1 ± 28.7	51.7 ± 29.1	54.2 ± 28.2	0.071
Previous kidney transplant	12 (12.5)	12 (14.1)	9 (9.7)	7 (8.6)	0.653
Admission systolic blood pressure	150 ± 23	156 ± 27	155 ± 28	152 ± 25	0.377
Admission systolic blood pressure < 100 mmHg	0 (0.0)	1 (1.2)	2 (2.2)	3 (3.7)	0.275
Discharged on alpha adrenergic medication	3 (3.1)	2 (2.4)	5 (5.4)	2 (2.5)	0.660
Standard DGF ^2^	7 (7.3)	30 (35.2)	66 (71.0)	60 (74.1)	<0.001

^1^ cPRA calculated by the DonorNet computer system using unacceptable values entered for a candidate. ^2^ Defined as dialysis within 1 week of kidney transplantation.

**Table 3 jcm-11-01535-t003:** Multivariate logistic regression model for 12-month eGFR <40 mL/min/1.73 m^2^.

Variable (Reference)	Odds Ratio (Confidence Interval)	*p* Value
DGF 4–10 Days (≤3 days)	2.11 (0.58–7.66)	0.137
DGF 11–20 Days (≤3 days)	1.58 (0.43–5.83)	0.371
DGF ≥ 21 Days (≤3 days)	5.73 (1.58–20.80)	<0.001
Donor Age (increasing)	1.04 (0.94–1.09)	0.023
Recipient Male (female)	3.02 (1.20–7.64)	0.002
Kidney Donor Profile Index (increasing)	1.00 (0.97–1.03)	0.674
Cold Ischemia Time > 30 h (≤30 h)	2.13 (0.80–5.65)	0.047
Donor Hypertension (none)	1.08 (0.42–2.78)	0.836
Donor Body Mass Index (increasing)	1.01 (0.97–1.05)	0.480
Donation After Circulatory Death (DBD)	1.18 (0.50–2.79)	0.613
Non-Mandatory Share Kidney (other)	0.77 (0.28–2.12)	0.512

DBD, donation after brain death.

**Table 4 jcm-11-01535-t004:** Transplant recipient secondary outcomes stratified by DGF duration.

Outcome *n* (%) or Mean *±* SD	DGF 0–3 *n* = 96	DGF 4–10 *n* = 85	DGF 11–20 *n* = 93	DGF ≥ 21 *n* = 81	*p* Value
12-month Patient Death	2/96 (2.1)	2/85 (2.4)	2/93 (2.2)	2/81 (2.5)	0.849
12-month Graft Failure	6/96 (6.3)	7/85 (8.2)	6/93 (6.5)	9/81 (11.1)	0.617
12-month Acute Rejection	13/91 (14.3)	7/79 (8.9)	11/89 (12.4)	13/76 (17.1)	0.830
12-month CMV viremia	8/88 (9.1)	14/78 (17.9)	12/82 (14.6)	10/70 (14.3)	0.420
12-month BK viremia	19/89 (21.3)	18/77 (23.4)	21/85 (24.7)	20/73 (27.4)	0.839

## Data Availability

The data presented in this study are available on request from the corresponding author.

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
