# Peer review of "Kidney Transplant Outcomes after Prolonged Delayed Graft Function"

_jcm, 2022, doi:10.3390/jcm11061535_

Round 1
Reviewer 1 Report
Donnelly et al, assessed the impact of Delayed Graft Function (DGF) on different kidney transplant endpoints such as 1-year eGFR as primary endpoint and patient survival, graft survival, acute rejection , CMV or BKV viremia as secondary endpoint in a monocentric cohort. As depicted in their introduction section, numerous reports already investigated this issue. Of course, all these reports faced limitations but I am not convinced the limitations of the present manuscript are less important.
Major comments:
-The authors chose to exclude patients with cPRA>99%. First, the authors must add cPRA abbreviation and the way it was calculated in the Material and Method section. I presume it was indicated as a percentage so the symbol '%' should be written accordingly.
-The authors claimed that recipients with cPRA>99% are excluded because their graft may be impacted by immunologic injury. Could the author provide any evidence of such higher immune injury? Did the biopsies of these recipients show more immune-related lesions as compared to low or medium cPRA? If not, this exclusion seems inappropriate.
-Instead of a bar graph with numbers, I strongly suggest the authors to present their Figure 1 as Bow and whiskers with median and IQ range.
-In Discussion section, the authors suggested that "These patients could be flagged for management strategies or future therapeutic interventions to reduce or prevent graft functional decline". Could the authors suggest one or two example(s) of such management strategies?
Author Response
Reviewer 1
Comment: The authors chose to exclude patients with cPRA>99%. First, the authors must add cPRA abbreviation and the way it was calculated in the Material and Method section. I presume it was indicated as a percentage so the symbol '%' should be written accordingly.
Response: We have added the “cPRA” abbreviation, its definition, and the % sign where it was inadvertently missing.
Comment: The authors claimed that recipients with cPRA>99% are excluded because their graft may be impacted by immunologic injury. Could the author provide any evidence of such higher immune injury? Did the biopsies of these recipients show more immune-related lesions as compared to low or medium cPRA? If not, this exclusion seems inappropriate.
Response: We did not include these highly sensitized recipients since most do not have DGF (as a result of receiving optimal kidneys at short cold ischemia times), but when they do have DGF, our QAPI metrics indicate that the cause of the DGF is generally immunologic. By excluding these patients, the study was methodologically positioned to assess outcomes of DGF that was more likely to be mediated by ischemia reperfusion injury or donor quality, rather than immunologic causes, thereby assessing the research question in somewhat more immunologically homogenous groups.
Comment: Instead of a bar graph with numbers, I strongly suggest the authors to present their Figure 1 as Bow and whiskers with median and IQ range.
Response: Since eGFR was normally distributed, we assessed mean and standard deviation rather than median and IQR.
Comment: In Discussion section, the authors suggested that "These patients could be flagged for management strategies or future therapeutic interventions to reduce or prevent graft functional decline". Could the authors suggest one or two example(s) of such management strategies?
Response: We have added that a management strategy would be to facilitate timely re-transplantation and previously indicated that thus far tested treatments have been shown to be effective in clinical trials.
Please see attachment.

Reviewer 2 Report
Donnelly, Keller, and Kayler proposed a retrospective evaluation about the impact of prolonged DGF in their internal cohort. However, some points should be more extensively discussed or even added.
- Despite some different definitions of DGF being adopted, most studies considered DGF patients all subjects, which necessitates a dialysis intervention in the first week of kidney transplantation. The use of sCr based evaluation could be included but should be compared to the “standard.” In this way, the Authors are invited to consider a sub-analysis clustering patient according to the most common definition and compare their results and significant variables between groups
- Many studies evaluated DGF, and some of them analyzed US data. Please expand and revaluate bibliography and discussion (including, for example, Bahl et al. Curr Opin Organ Transplant 2018; Tapiawala SN et al. J Am Soc Nephrol 2010; Zens TJ et al. Clinical Transplantation 2018; Lee J et al. Sci Rep 2017)
- The role of hypotension and drug therapies in DGF determination is crucial (see Dolla et al. Plos One 2021); despite the Authors mentioning the absence of these parameters in their limits, some data are strictly needed (i.e., the number of patients with hypotension during the surgery and immediately after transplant; the difference in induction therapies or immunosuppressant through level during the first week)
Author Response
Reviewer 2
Comment: Despite some different definitions of DGF being adopted, most studies considered DGF patients all subjects, which necessitates a dialysis intervention in the first week of kidney transplantation. The use of sCr based evaluation could be included but should be compared to the “standard.” In this way, the Authors are invited to consider a sub-analysis clustering patient according to the most common definition and compare their results and significant variables between groups.
Response: We did not use the standard definition of DGF since this would result in the removal of both pre-emptive patients and those without DGF, resulting in a sample size that lacks power to draw meaningful conclusions.
Comment: Many studies evaluated DGF, and some of them analyzed US data. Please expand and revaluate bibliography and discussion (including, for example, Bahl et al. Curr Opin Organ Transplant 2018; Tapiawala SN et al. J Am Soc Nephrol 2010; Zens TJ et al. Clinical Transplantation 2018; Lee J et al. Sci Rep 2017)
Response: We have reviewed these studies and incorporated where applicable.
Comment: The role of hypotension and drug therapies in DGF determination is crucial (see Dolla et al. Plos One 2021); despite the Authors mentioning the absence of these parameters in their limits, some data are strictly needed (i.e., the number of patients with hypotension during the surgery and immediately after transplant; the difference in induction therapies or immunosuppressant through level during the first week)
Response: We have indicated that a very small minority of our patients did not receive ATG (n=4). Therefore, induction as a potential confounder was not added to the table for comparison purposes. To assess the role of hypotension we had ascertained admission systolic blood pressure and discharge on alpha adrenergic medication as shown in Table 2. All of these metrics for assessing the adequacy of perfusion to the kidney have limitations including hypotension during surgery and immediately after transplant.
We appreciate the opportunity to respond to the reviewers’ criticisms and hope that these modifications to our manuscript will render it acceptable for publication in Journal of Clinical Medicine. I look forward to hearing from you shortly.
Sincerely,
Please see attachment.

Round 2
Reviewer 2 Report
Despite appreciating the authors' comments, some questions remain unanswered.
1) The standard definition of DGF does not necessarily cause the "removal" of pre-emptive patients (that could thereby need dialysis in case of impaired graft function) and those without DGF (that could be maintained as an internal control group, with subsequent analysis). I agree that standard definition already has limitations, and pre-emptive patients could theoretically be excluded in part, but a more detailed comment must be included in the text.
2) No previous suggested studies for DGF and hypotension have been discussed or included. Please, reevaluate and comment appropriately.
3) The absence of immunosuppressant through level during the first week should be included in the limitation.
Author Response
Reviewer
Comment: The standard definition of DGF does not necessarily cause the "removal" of pre-emptive patients (that could thereby need dialysis in case of impaired graft function) and those without DGF (that could be maintained as an internal control group, with subsequent analysis). I agree that standard definition already has limitations, and pre-emptive patients could theoretically be excluded in part, but a more detailed comment must be included in the text.
Response: We have updated Table 2 to include standard DGF frequencies. Additionally, in the limitations section, we specifically emphasize that there are numerous definitions of DGF and DGF duration, and a time-based analysis of serum creatinine change has been suggested to perform best in being able to predict reduced graft function at one year.
Comment: No previous suggested studies for DGF and hypotension have been discussed or included. Please, reevaluate and comment appropriately.
Response: Hypotension was included as a baseline recipient variable to be as complete as possible; however, the frequency of this potential confounder was similar across groups and was too low to be of any value in interpreting our results. Therefore, in the limitations section where we discuss this limitation, we have also referenced a paper about DGF and hypotension.
Comment: The absence of immunosuppressant through level during the first week should be included in the limitation.
Response: This concept has been added to the limitation section.
We appreciate the opportunity to respond to these comments and hope that our modifications to the manuscript will render it acceptable for publication in Journal of Clinical Medicine.
